# A Robustness Evaluation of Machine Learning Algorithms for ECG Myocardial Infarction Detection

**DOI:** 10.3390/jcm11174935

**Published:** 2022-08-23

**Authors:** Mohamed Sraitih, Younes Jabrane, Amir Hajjam El Hassani

**Affiliations:** 1MSC Laboratory, Cadi Ayyad University, 40000 Marrakech, Morocco; 2Nanomedicine Imagery & Therapeutics Laboratory, EA4662—UBFC, UTBM, 90000 Belfort, France

**Keywords:** ECG myocardial infarction classification, support vector machine (SVM), k-nearest neighbors (KNN), random forest (RF), model robustness, inter-patient paradigm

## Abstract

An automatic electrocardiogram (ECG) myocardial infarction detection system needs to satisfy several requirements to be efficient in real-world practice. These requirements, such as reliability, less complexity, and high performance in decision-making, remain very important in a realistic clinical environment. In this study, we investigated an automatic ECG myocardial infarction detection system and presented a new approach to evaluate its robustness and durability performance in classifying the myocardial infarction (with no feature extraction) under different noise types. We employed three well-known supervised machine learning models: support vector machine (SVM), k-nearest neighbors (KNN), and random forest (RF), and tested the performance and robustness of these techniques in classifying normal (NOR) and myocardial infarction (MI) using real ECG records from the PTB database after normalization and segmentation of the data, with a suggested inter-patient paradigm separation as well as noise from the MIT-BIH noise stress test database (NSTDB). Finally, we measured four metrics: accuracy, precision, recall, and F1-score. The simulation revealed that all of the models performed well, with values of over 0.50 at lower SNR levels, in terms of all the metrics investigated against different types of noise, indicating that they are encouraging and acceptable under extreme noise situations are are thus considered sustainable and robust models for specific forms of noise. All of the methods tested could be used as ECG myocardial infarction detection tools in real-world practice under challenging circumstances.

## 1. Introduction

Researchers have (in many ways) investigated artificial intelligence in medicine to boost the efficiency and effectiveness of medical diagnoses and in healthcare, which can be very helpful to doctors and medical staff. However, given a variety of factors, such as illnesses that require extensive analyses and diagnoses using expensive machines, artificial intelligence could be used as a key tool to assist doctors concerning a variety of diagnoses. Globally, the leading cause of death is cardiovascular disease (CVD), which is a set of disorders of the heart and blood vessels. In 2019, about 17.9 million people died from CVDs, representing 32% of all deaths in the world. Among these deaths, 85% were because of heart attacks and strokes [1]. CVDs refer to all illnesses related to the heart and vessels, including coronary heart disease, cerebrovascular disease, peripheral arterial disease, rheumatic heart disease, congenital heart disease, deep vein thrombosis, pulmonary embolism, strokes, and heart attacks. Heart attacks, also often called myocardial infarction (MI), are a result of a partial or total blockage of coronary arteries due to the buildup of plaque, which causes a shortage in the blood supply to the heart muscle (myocardium) [2]. These blocks severely reduce blood supply to the myocardium. When the flow of oxygen-rich blood is inadequate, cardiac muscle damage develops and starts dying if blood flow to the heart is not restored in time [3]. A heart attack occurs when there is a full blockage of blood flow because a segment of the heart muscle is destroyed [4]. Thus, it is the most prevalent and deadly cardiovascular disease, causing more than 25% of all fatalities worldwide, according to the WHO Liu et al. [5].

As a result, quick and proper identification of MI is critical to enhancing life expectancy and improving a patient’s quality of life. In terms of diagnosis, among all the existing methods, an MI patient can be diagnosed by an electrocardiogram (ECG) [6,7], echocardiography [8], magnetic resonance imaging (MRI) [9], and other methods [10,11]. For patients with critical conditions, the initial MI diagnostic tool is the ECG, as it is a non-invasive and economical primary tool that is frequently available inside ambulances and emergency rooms and can be used to identify cardiac abnormalities [12]. The ECG signal provides critical information about the heart’s function and rhythm, and as the heart suffers from episodes of ischemia, infarction, arrhythmia, morphological alterations in the ECG signal are directly reflected as irregularities in waves or rhythm [13,14]. Furthermore, when used and interpreted appropriately, the ECG provides a quick and highly reliable diagnosis of heart failure [15]. Thus, dynamic alterations in the electrocardiogram (ECG), such as ST-segment depression and elevation, T wave inversion, and pathological Q waves, help in the diagnosis of myocardial infarction [16,17]. The hand-operated interpretation of ECG recordings [18] can take a significant amount of time, resulting in inter-observer variability [19,20]. In this case, a computer-aid system can be very helpful to speed up this process and help doctors provide more accurate medical decisions [19,21]. A substantial amount of research has been dedicated to automated systems for biomedical ECG signal interpretation for the early detection of myocardial Infarction.

## 2. Literature Overview

The recent research on ECG classification computer-aid systems is based on new techniques of artificial intelligence. Various types of these techniques have been investigated and analyzed for this purpose and other purposes [22,23], such as decision trees [24], random forest (RF) [25,26,27], support vector machine (SVM) [28,29], k-nearest neighbor (KNN) [30], the hybrid FFPSONeural network classifier [31], in addition to other methods, such as [32,33,34]. In [35], the authors incorporated two categories, normal and MI, into their investigation. They preprocessed the signal and retrieved features such as R peaks with a detection algorithm and then ranked the generated features with a *t*-test. The data were fed to classifiers such as k-nearest neighbor (KNN), support vector machine (SVM), probabilistic neural network (PNN), and decision tree (DT) to determine the normal and MI classes. The SVM classifier produced significant results, with an accuracy of 97.96%. In [36], the authors investigated the prognostic effectiveness of machine learning for major adverse cardiovascular events (MACEs) in individuals with acute myocardial infarction. They employed six machine learning models in their study to assess 24 clinical variables (chosen by logistic regression analysis) in 500 patients with AMI to identify MACEs. With an accuracy of 0.820, the random forest (RDF) model performed the best. This paper [37] introduces a novel two-band optimal biorthogonal filter bank (FB) for ECG signal processing. The authors employed the newly created wavelet FB to decompose the ECG signal into six subbands (SBs) and then performed three discriminating features on all six SBs, namely fuzzy entropy (FE), signal-fractal-dimensions (SFD), and Rényi entropy (RE). The resulting characteristics are given in the k-nearest neighbor algorithm (KNN) to differentiate between classes: normal and MI. Using the 10-fold cross-validation (CV) approach, the proposed system achieved an accuracy of 99.74%. Several other methods for detecting myocardial infarction (MI) using the inter-patient paradigm have been developed, including [5,38,39,40,41]. The authors in [40] proposed a unique method that involved fusing energy entropy and morphological features for MI detection using the SVM model via a 12-lead ECG. The method is based on the maximal overlap discrete wavelet packet transform (MODWPT) to extract different types of features and it uses three methods to reduce the computational complexity and redundant information. Their method achieved good performance results with an accuracy of 92.69% for the inter-patient paradigm. The authors in [39] introduced an application of a support vector machine (SVM) and a genetic algorithm (GA) to classify electrocardiogram (ECG) signals of myocardial samples from normal. The authors used ECG records from the database (PTBDB) and applied a feature extraction of the ECG morphological, time domain, and discrete wavelet transform (DWT) of 148 MI and 52 normal ECG recordings from the database. The authors achieved a sensitivity and specificity of 87.80% and 85.97%, respectively. According to our research, there have been few or almost no studies in recent years that evaluated ECG signal classification model robustness. In this study [42], the authors employed a deep learning approach—a convolutional neural network—to distinguish three classes (normal, AF, and STD) by applying transformed ECGs to images (attractor reconstruction, etc.). They used a random SNR between 5 and 10 dB to assess the durability of their classifier trained on clean data against three forms of noises (em, bw, and ma). The F1-scores for the scalogram transforms and the SPAR attractor on the clean ECG data set were 0.70 and 0.79, respectively. However, once the model trained on clean data was applied to identify the classes from the noisy data sets, F1-scores dropped by (up to) 0.18. In this work [43], the authors constructed a CNN for the classification of 12-lead ECG signals, and they used three defense systems to increase the robustness of the CNN classifier for ECG classification against a PGD-100 attack and white noise. The evaluation study reveals that the modified CNN achieved an acceptable F1-score and average accuracy, and the defense methods improved their CNN’s robustness against adversarial noise and white noise, with a small decrease in accuracy on clean data. However, when the noisy test data set of the 100-PGD attack was evaluated at different levels of SNR, the accuracy of the best model dropped significantly to less than 0.5 for noise levels greater than 0.01 and 0 for high SNR levels. Moreover, for the white noise, the best classifier achieved a high accuracy of around 0.83 and dropped to a minimum accuracy of around 0.65 at high SNR levels. Both of these studies used intra-patient paradigms, which gave overoptimistic estimates of the actual classifier performances [23].

In this study, we investigated and present a new approach to evaluate the robustness and durability of an automatic ECG myocardial infarction detection system to classify the myocardial infarction against data infected with a different variety of noises, performing no feature extraction. We employed three well-known supervised machine learning models: support vector machine (SVM), k-nearest neighbors (KNN), and random forest (RF). We tested the performance and robustness of these techniques in classifying normal (NOR) and myocardial infarction (MI) ECG signals using real ECG records from the PTB database after normalization and segmentation of the data by performing an inter-patient paradigm separation and noise from the MIT-BIH noise stress test database (NSTDB), and we measured four metrics: accuracy, precision, recall, and F1-score. Figure 1 illustrates a summary of the overall procedures used in this paper.

The rest of this paper is organized as follows: in the next section, we present the materials and methods. Section 4 describes the results; a detailed discussion of these results is presented in Section 5. Finally, we conclude the paper in Section 6.

## 3. Materials and Methods

### 3.1. Support Vector Machines

Because of its outstanding generalization performance in supervised machine learning, the support vector machine (SVM) is one of the most widely used classification techniques for solving binary classification problems [44,45]. The way SVM works for classification tasks is by constructing a hyperplane in a multidimensional space that is dedicated to separate particular class labels. The multidimensional space depends on the number of features used as input. The idea behind creating the hyperplane is to separate the classes by maximizing the geometric margin between the input data classes mapped in a higher-dimensional space and minimizing the empirical classification error [46,47]. This process of classification is performed using the kernel functions [48]; in both cases, the linear or nonlinear classifier. We can apply SVM as a linear classifier by using a linear kernel function or as a non-linear classifier by using polynomial and sigmoid kernel functions. However, for any classification task, the choice of the kernel function remains a challenging task. To demonstrate how the decision function works mathematically, we will consider a training set of N samples ($y_{j}$, $x_{j}$), *j* = 1, ... , N, where $x_{j}\in {I\;R}^{n}$ represents the multi-dimensional feature vector of the *j*th example and $y_{j}\in I\;R$ represents the matching class label as $y_{j}\in \pm 1$. The g(x) function, which refers to the decision function learned from the training set, reveals a representation of the optimal hyperplane that will predict the class label in subsequent tests. The decision function with the kernel is defined as follows [49,50]:(1)$$g\left(x\right)=\mathrm{sign}\left(\sum_{i\in SV_{s}}\alpha_{i}y_{i}K\left(x,x_{i}\right)+b\right)$$
where α is the Lagrange multiplier for each training data set and $K(x_{j},x)$ is the kernel function that graphs the inputs into a larger dimensional space, represented as in the state of the polynomial Kernel [49]:(2)$$K\left(x,x_{i}\right)=\sum {\left(x,x_{i}\right)}^{d}$$
where *d* is the degree of the polynomial function.

In the literature, SVM is one of the most widely used algorithms in biology [51] particularly for ECG arrhythmia detection [44,52,53].

### 3.2. K-Nearest Neighbors

The K-nearest neighbors (KNN) algorithm is one of the most frequently used supervised machine learning methods, especially for classification tasks, because of its simplicity to comprehend and diagnose. However, it is known as a non-parametric lazy algorithm, as opposed to the other approaches, which utilize different methods to fit the data for training. There is no training step with the KNN algorithm. Instead, it conducts the computation at the time of classification. The KNN uses the labels of the nearest training samples in the feature space to classify the feature vectors. By calculating the distance (such as the Hamming, Euclidean, Minkowski, etc.) between an unidentified feature vector or new instance and all vectors in the training set, the k-nearest neighbor is determined. The neighbor’s vote helps to assign the unknown feature vector to the class to which the closest k samples mostly belong, and it takes the class with the most votes as a prediction [54,55]. The KNN has been widely used and employed in some recent ECG classification studies [21,56,57,58,59].

### 3.3. Random Forest

The random forest (RF) algorithm is also a widely used method for supervised machine learning problems, especially for classification; it was first proposed by Breiman [60]. The RF algorithm is essentially an ensemble of decision trees used to train and predict the results. It is considered a parametric algorithm regarding the number of trees in the forest and a stochastic method because of its two origins of randomness—the random attribute sub-set selection and the bootstrap data sampling. This randomness prevents over-fitting during the training process. The construction of an RF model depends on various parameters, which are: the number of trees, which is considered very important, and the maximum depths and splits. The internal process of choosing a split by the decision trees consists of picking their splitting properties from a random subset of k characteristics at each internal node. Then, the best split is taken within these randomly chosen attributes, and it builds the trees without trimming. Several applications have used the RF method for classification in many challenges, particularly in areas with larger numbers of attributes and situations, because of its high-speed, outstanding prediction accuracy, suitable tolerance for outliers and noise, and resistance to overfitting [21,61,62].

### 3.4. Performance Evaluation Measures

To evaluate the classification results, many performance metrics are available. The four most commonly used metrics in the literature are [63]:

*Accuracy*: may be defined as the accurate classification ratio of the total categorized outcomes.
(3)$$\mathrm{Accuracy}=\frac{TP+TN}{TP+TN+FP+FN}$$*Precision*: positive predictive and is often known as the ratio of real positives to the total number of positively-predicted samples; it is defined as:
(4)$$\mathrm{Precision}=\frac{TP}{TP+FP}$$*Recall*: also known as sensitivity—it is the ratio of positively-predicted samples to the total number of really positive samples; it is defined as:
(5)$$\mathrm{Recall}=\frac{TP}{TP+FN}$$*f1-score*: the harmonic mean of precision and recall; it is expressed as:

(6)$$f1\text{-}\mathrm{score}=2\times \frac{\mathrm{Precision}\times \mathrm{Recall}}{\mathrm{Precision}+\mathrm{Recall}}$$
where these parameters are: *TP* = true positive(s) (correctly classified labels), *FN* = false negative(s) (unclassified labels), *TN* = true negative(s) (correctly unclassified labels), and *FP* = false positives() (incorrectly classified labels).

### 3.5. Signal Quality Evaluation Measures

In this study, we used two different approaches to measure the quality of the ECG signal samples. The first was the mean square error (MSE), which measures the difference between the original and filtered ECG signals; it is defined as follows:(7)$$MSE=\frac{1}{L}\sum_{k}^{L}{\left(ECG_{\mathrm{original}}^{k}-ECG_{\mathrm{filtred}}^{k}\right)}^{2}$$
where *L* is the length of the signal.

The second evaluation measurements we performed were approximate entropy, which is recommended for a short and noisy data set analysis [64], and fuzzy entropy [65,66] of the noisy signals at different levels of SNR. In Grassberger and Procaccia [67], Eckmann and Ruelle [68], the authors developed the approximate entropy (ApEn). It enables a statistical metric to quantify the complexity of fluctuation predictions within a time series, such as an ECG signal. A higher value of approximation entropy reflects less regularity of patterns within the time series, indicating poor signal quality, and vice versa. An N-dimensional $v_{n}$ time series is subdivided into m-dimensional $u_{m}\left(i\right)$ vector sets. The number of vectors that are as close together as the Euclidean $n_{i}^{m}\left(r\right)$ is given as $d[u_{m}\left(i\right),u_{m}\left(j\right)]⩽r$, where r is the tolerance; this number is used to calculate the probability:(8)$$C_{i}^{m}\left(r\right)=\frac{n_{i}^{m}}{\left(N-m+1\right)}.$$

The approximate entropy is given by [64]:(9)$$ApEn\left(m,r\right)=\lim_{N\to \infty }\left[\Phi^{m}\left(r\right)-\Phi^{m+1}\left(r\right)\right],$$
where $\Phi^{m}\left(r\right)=\frac{1}{\left(N-m+1\right)}\sum_{i=1}^{N-m+1}\ln C_{i}^{m}\left(r\right)$.

In Chen et al. [69], the authors proposed the fuzzy entropy used in the complexity computation of the EMG signal. The similarity definition of vectors in approximation entropy is based on the Heaviside function, which is defined as:$$\theta =\left\{\begin{array}{cc} 1 & \;\mathrm{if}\;z⩾r\\ 0 & \;\mathrm{if}\;z<r. \end{array}\right.$$

Instead of this similarity function, for the fuzzy entropy, a similarity degree was calculated by a fuzzy function $\mu (d_{ij}^{m},n,r)$, which is an exponential function, described as:(10)$$D_{ij}^{m}=\mu \left(d_{ij}^{m},n,r\right)=\exp\left(-\frac{{\left(d_{ij}^{m}\right)}^{n}}{r}\right).$$

The fuzzy entropy is defined as [69]:(11)$$FzEn\left(m,n,r\right)=\lim_{N\to \infty }\left[\ln\Phi^{m}\left(n,r\right)-\ln\Phi^{m+1}\left(n,r\right)\right],$$
where $\Phi^{m}\left(n,r\right)=\frac{1}{\left(N-m\right)}\sum_{i=1}^{N-m+1}\left(\frac{1}{\left(N-m-1\right)}\sum_{j=1,j\neq i}^{N-m+1}D_{ij}^{m}\right)$.

### 3.6. ECG Databases

We used real ECG signals from the PTB-ECG database, which involve ECG data obtained from Germany’s National Metrology Institute, Physikalisch Technische Bundesanstalt (PTB) [70]. The database contains 549 records from 290 subjects, comprising healthy patients and other patients who suffer from heart diseases. Each person is represented by a different number of recordings, from one to five records, and each record is composed of 15 leads, which are 12 standard leads (I, II, III, aVR, aVL, aVF, V1, V2, V3, V4, V5, V6) and three Frank lead ECGs (Vx, Vy, Vz). The signals are sampled at 1000 Hz per second, with a resolution of 0.5 μV and they have variable durations. The second database we used was the MIT-BIH noise stress test database (NSTDB) [71]. This collection contains 12 half-hour ECG recordings as well as 3 half-hour noise recordings typical of ambulatory ECG recordings.

### 3.7. Data Preparation

In our experiment, we used 106 patients in total from the PTB-ECG database, which included 53 healthy patients labeled as NOR, which means normal, and 53 patients suffering from myocardial infarction, labeled as MI. By performing the inter-patient paradigm separation, we divided the patients into two groups: G1 contained twenty-seven subjects, and G2 contained twenty-six subjects of each class (NOR and MI). For this separation, we used one record per patient, and each record contained 12 standard leads. Table 1 illustrates the format of the data separation.

The first group G1 was used for training the models and contained the data of healthy patients: (104, 105, 116, 117, 121, 122, 131, 150, 155, 156, 165, 166, 169, 170, 172, 173, 174, 180, 182, 184, 185, 198, 214, 229, 233, 234, 235) and patients who had myocardial infarction: (001,002, 003, 004, 005, 006, 007, 008, 009, 010, 011, 012, 013, 014, 015, 016, 017, 018, 019, 020, 021, 022, 138, 140, 152, 163, and 193). The second group, G2, was divided into a validation and test set by 20% and 80%, respectively, and contained data from healthy subjects: (236, 237, 238, 239, 240, 241, 242, 243, 244, 245, 246, 247,248, 251, 252, 255, 260, 263, 264, 266, 267, 276, 277, 279, 281 and 284), as well as data from patients with myocardial infarction: (023, 024, 025, 026, 027, 028, 029, 030, 031, 032, 033, 034, 035, 036, 037, 038, 039, 040, 041, 042, 043, 044, 045, 046, 047, 048).

### 3.8. Data Pre-Processing

#### 3.8.1. Normalization

After the separation process, we used 12 leads of each record extracted from the database and preprocessed 32,000 samples of each lead’s record. The preprocessing of the data samples consisted of normalization of each lead’s ECG signal between 0 and 1, using the min–max normalization. This method adjusts the value of the min limit (a) and max limit (b) on the amplitude of a signal to the needed range with the assurance of no change in the shape or pattern of the signal characters. The mathematical function of the normalization with min–max normalization is as follows:(12)$$x{}^{\prime }=a+\frac{\left(x-x_{min}\right)\left(b-a\right)}{x_{max}-x_{min}}$$
where $x{}^{\prime }$ is the normalized signal, $x_{max}$ and $x_{min}$ are the maximum and minimum values of the data set, respectively, *a* and *b* are the min-limit and max-limit values. In this study, the values chosen in the preprocessing were 0 and 1, respectively. After normalization, we used a low-pass Butterworth filter with a filter order of N = 8 and a cutoff frequency of $f_{c}=0.04$ to remove noise from the data.

#### 3.8.2. Segmentation

After the filtering process, we performed a segmentation sampling of a 0.65 s segment for each beat, which meant 650 samples in each segment, as can be seen in Figure 2. The segment’s section was divided into two intervals of $t_{1}=0.30$ s before and $t_{2}=0.35$ s after the R peak position in the signal. To preserve the data fluctuations and improve the understanding of the signal properties, we subtracted a linear trend from the generated data by computing the least-squares regression line to determine the growth rate r. The differences (i.e., variations from the least-squares fit line) were then subtracted from the data. Using this strategy, the model may stay focused more on the class characteristics during training. The last results are illustrated in Table 2.

### 3.9. Hyperparameters Selection

In any machine learning model, there are multiple parameter keys to be identified, which are referred to as hyperparameters. When optimizing for performance, hyperparameters assist in regulating the behavior of machine learning algorithms in order to strike the proper balance between bias and variance. However, there are no easy–fast rules that ensure optimal performances on specific data sets. The traditional method for determining the model parameters that can obtain the best prediction outcomes is to run multiple types of hyperparameters on each model, which is a time-consuming operation. We conducted a grid search on each model in this study by supplying a mix of parameter grids. Table 3 depicts the range grid of the hyperparameters adopted in our study.

To identify the hyperparameters, we used the training set for training the models, while the validation set was considered a testing set to validate the model performance. The halving grid search fit the parameter grid on the training set and checked its behavior on the validation set, employing successive halving for each model hyperparameter combination and each train–validation split. The successive halving is a repetitive selection procedure in which all the parameter combinations are evaluated with a limited number of resources at the first iteration. Only some options are picked for the next iterations, and so on. Therefore, only a subset of options survives until the last iteration, implying that they are typically ranked among the top-scoring options in all iterations. The best estimator is chosen by taking the average of the top-performing models. The complete process is performed by the scikit-learn [72] packages, using the HalvingGridSearchCV tool with a factor of 3 (the rate at which the resources grow and the number of candidates decreases). This experiment yielded the appropriate set of hyperparameter combinations that helped the model provide the most accurate predictions. The final set of hyperparameter combinations that will be applied in the remaining experiments of this study are as follows:SVM: C = 1, gamma = 0.1, kernel = ‘rbf’.KNN: metric = ‘euclidean’, n-neighbors = 5, weights = ‘distance’.RF: n-estimators = 800, max-depth = 25, min-samples-split = 5.

## 4. Results

### 4.1. Model Tuning

In the next step, we evaluated the hyperparameters revealed in the grid search to express the behavior over time of each model with the hyperparameters chosen and to show different changes that can accrue and prevent over-fit and under-fit. To estimate how the models are supposed to behave when employed to produce predictions on unseen data during training and to know if they are good-fitted models, we investigated each model with its hyperparameters by making a dual learning curve of the training set and a validation set with a repeat of five times, measuring the accuracy of each one. The final results are the means over these repetitions. The learning curve reflects the model’s training process by evaluating the accuracy for every size of training and validation set, ranging from small to large data sets. As more data are fitted, the accuracies must increase, and if they hit 1 (100% accuracy), the training data set is perfectly learned. The learning curve generated from the training set reveals whether the model is learning, but the learning curve computed from the hold-out validation set indicates whether the classifier is generalizing unseen data samples. As we can see from the training set’s accuracy in Figure 3, which exceeds 0.99, all classifiers achieved nearly perfect training, except for the SVM model, which achieved just a maximum of 0.85. However, the models delivered promising predictions, as evidenced by the validation set’s accuracy, which increases as more data are fitted, allowing the model to learn more features and produce consistent predictions. The highest level of accuracy attained was 0.75, except for the KNN, which reached a maximum of 0.70. These outcomes are not surprising given that we applied an inter-patient paradigm, which implies that the data used to train the model differs from those used to evaluate it, taking into account the inter-individual differences. As a result, it indicates that all the models are well-fit and generalize better on previously unseen data.

Following the process of training the models and evaluating them using a validation set, the next step was to analyze all of the models on the unseen data set, which is the test set, in order to predict the classes.

As a form of performance evaluation, we measured four metrics—precision, recall, and F1-score, for each class of all the models, and the last metric was the accuracy of the models in predicting all the classes, as illustrated in Table 4. Considering the first three metrics, RF obtained the best precision and F1-score in predicting both classes and the highest recall for the MI class. On the other hand, SVM achieved the highest recall as well as KNN for the NOR class. Thus, in terms of comparing the model’s performance (with regard to accuracy), RF took the lead, achieving the best results in predicting the classes of the unseen data by achieving an accuracy of 0.74. Overall, the performances of all the models in detecting MI were low (with regard to detecting NOR samples). However, the results are quite encouraging since the models identified both classes from an unseen data set in which the same patient’s beats were present at one stage, either during the training or testing phase. To provide an unobstructed view of the number of samples categorized into one of two classes by each model, we present each model’s classification report as illustrated in Table 5.

All the models provided nearly the same results, as we can see from Table 5, with little variation. Unlike the RF model, which performed the best in predicting the MI class, SVM and KNN obtained promising predictions, with good results for the NOR class. Regardless, the two classes (NOR and MI) seemed difficult to predict since all the models mislabeled a significant number of samples. The KNN misclassified 0.39 of all samples as MI, which is deemed a significant number when compared to RF, as it just misclassified 0.27 of all the samples. For the NOR class, all the models misclassified almost the same number of samples. Moreover, it is sometimes challenging for the model to predict all of the classes perfectly, even if the data are suitable, especially with inter-patient separation; this issue can occur if the training data does not involve all the features of a certain class. However, it is reasonable to expect some misclassification of some samples, with a small variance.

### 4.2. Model’s Robustness Test

We investigated the robustness of the trained models in predicting the classes against a range of real-world noises using an infected test data set. We conducted two experiments for this objective: The first one was to test the trained models on the same data test set as before, but this time without the filtering process. The second was to run the trained model on the same data test set as before, but this time with a variety of added noises. For the first experiment, we used the identical preparation steps as before (except for the filtering) to maintain the signal, as it was imported from the database; we separated the data as above, and then tested all the models on this changed test data set. Figure 4 illustrates a sample of the two classes.

As part of the performance evaluation, we measured the same metrics, in the same manner, to analyze the performances of the trained classifiers in predicting the two classes and we compared them to the earlier achieved results. Table 6 exhibits the results. From the first point of view, it is obvious that the performances of the models were deteriorating. In terms of the first three metrics, SVM and KNN F1-scores decreased by the same amount for both classes, as did precision and recall, but with minor differences of 0.01–0.02. Compared to the other models, the RF model declined dramatically, dropping by 0.12 in precision for the MI class and by 0.16 and 0.11 in recall and the F1-score, respectively, for the NOR class. Furthermore, unlike SVM and KNN, which declined by 0.06 and 0.04 points, respectively, the RF model dropped dramatically by 0.09, which was unexpected given that it attained the greatest accuracy in comparisons to the same models in the previous experiment, Table 4. As a consequence, while dealing with the same changed test data set, the SVM outperforms other models as a solid classifier. However, the model’s performance was expected to decline because, first, the data were altered, and second, each model employed a different structured approach and algorithm to learn the features of the data set fitted.

Next, we provided a classification summary for each classifier as an extra explanation of the classifier’s performance, which provides an accurate perspective of the classes that are correctly predicted and the classes that are incorrectly predicted.

As shown in Table 7 of the classification report, each classifier incorrectly forecasted a specific number of samples fed to be predicted. The KNN classifier comes out on top by predicting 40% of the MI samples as NOR, which appears to be a very large misclassification. In the same manner, the SVM classifier performed as well, with 34% of the misclassification, as did the RF, with 27% of the misclassified samples. However, in comparison to the other models, this appears to be a tolerable misclassification for the RF mode. Despite that, the RF classifier, on the other hand, takes a great part in misclassifying the NOR samples as MI, which appears to be a somewhat significant misclassification. Similarly, the KNN classifier performed as well, with 31%, followed by the SVM with 30% of misclassified samples. SVM and KNN models predict the NOR class better than the RF classifier, which predicts the MI class better. Overall, their performances seem acceptable.

In the second part of this experiment, we chose actual noises from the MIT-BIH noise stress test database (NSTDB) to test the robustness of the models against these noises. We added three realistic noises, one at a time, from this data set to the above-mentioned preprocessed and filtered testing data set: baseline wander, muscle (EMG) artifact, and electrode motion artifact. We applied these noises to each ECG signal lead’s record for simulation, using five degrees of a signal-to-noise ratio (SNR) from best to worst signal quality: 16 dB, 8 dB, 2 dB, and −2 dB. The final results of the ECG sample are illustrated in Figure 5.

### 4.3. Quality Evaluation of the Noisy Signals

Before feeding the modified test data set to the classifiers, we evaluated the quality of the modified test data set ECG signal samples (after the various noises were added, to illustrate the quality of the ECG samples to be fed to the classifiers to predict the MI and the NOR classes). For this reason, we performed two different measurements. The first measurement was the analytical mean square error (MSE), which measured the difference between the original and filtered ECG signals. However, in our case, we measured the difference between the above-mentioned, preprocessed, and filtered test data sets as the original signals and the same test data set after adding the various noises, one at a time.

The results shown in Figure 6 demonstrate that MSE increased as SNR levels increased for both classes of signal samples; this is not surprising given that we purposely degraded the ECG signal samples by adding noise. This result shows how much the signal differs after adding various forms of noise. However, certain types of noise appear to have considerable impacts on the signal. As we can see, the most harmful noise that destroyed the NOR signal samples appeared to be the muscle artifact, followed by the baseline wander. Similarly, the most damaging noise for the MI class appeared to be the electrode motion artifact, followed by the muscle artifact noise. The features and characteristics of the class signal samples could be very challenging for the classifiers to predict as the SNR of the added noise decreases, which means a poor quality signal. The approximate entropy and fuzzy entropy were the second assessment metrics that we used. In this experiment, we set the embedding dimension m to 2 and the tolerance r to 0.2 for both the approximate entropy (ApEn) and the fuzzy entropy (FuEn). We set the gradient of the boundary n to 2.

Figure 7 and Figure 8 show the ApEn and FzEn measurements after various forms of noise were added. These measurements estimated the level of complexity of fluctuations within a signal. According to the findings, the approximate and fuzzy entropy values increased with respect to the SNR levels, implying that all noises caused a significant impact on the class samples, resulting in a lower regularity of the patterns within the signal. However, certain noises had a greater impact on the signal than others. The baseline wander had the most significant impact on the signal, achieving the highest ApEn and FzEn at all SNR levels, which means a much lower regularity of the MI sample patterns, followed by the muscle artifact. The muscle artifact, on the other hand, had a greater impact on the NOR samples, achieving the highest ApEn and FzEn at all SNR levels, followed by the baseline wander. In both classes, the electron motion artifact had little effect. Overall, all the noises altered the pattern’s irregularity and predictability for both classes, making it extremely difficult for classifiers to properly predict the maximum number of samples.

The last experiment we performed was to evaluate the trained classifiers against this changed test data set after ensuring that the signals of the class samples had changed in terms of the regularity and predictability of patterns when different noises with various levels of SNR were added, which would be difficult for the classifiers to predict. We fed this changed test data set to the trained classifiers; it is worth noting that the trained classifiers were the same ones used in the previous experiments; we assessed the four metrics (as conducted above) to obtain a clear view of the performance comparison.

Figure 9 illustrates the results of the precision metric measured from each model in predicting both classes in each test data set added with noise. For the test data set with added muscle artifact noise, the models maintained high precision in predicting the classes. As we can see, at an SNR level of 16 dB, all the models achieved a precision above 0.65 in predicting the NOR class and 0.60 for the MI class, with the highest precision achieved by the RF, followed by SVM and KNN, respectively, for the NOR class. On the other hand, for the MI class at SNR = 16 dB, each model performed differently, but the performance shifted from an SNR of 8 to −2 dB, with the RF taking the lead, followed by the SVM and KNN, respectively. However, interestingly, for the class NOR, the models conserved high-precision performances above approximately 0.50 as the quality of the signals decreased, unlike for the MI class, which decreased at SNR = −2 dB, achieving a precision of under 0.50. Similar results were obtained when predicting the NOR class for the test data set with additional electrode motion noise, except for the MI class, where KNN took the lead at SNR = 16 dB, followed by the RF and SVM, which had identical performances, but shifted at SNR = 8 dB, giving the lead to the RF and SVM. In terms of performance decreasing, the results for the test data set with the added baseline wander noise were essentially the same for both classes. With minor differences, all the models obtained precisions above 0.53 for the NOR class and above 0.47 for the MI class at SNR = −2 dB, with the RF taking the lead, followed by the SVM and KNN, respectively, except for the level of SNR = −2 dB, where the KNN altered and the SVM dropped. Overall, for both classes, the RF model outperformed the other models in terms of high-precision performance, with a small variance as signal quality decreased.

We describe the recall metric achieved by each model in predicting both classes in each test data set with added noise in Figure 10. Considering all the noises added, the results seem interesting. The models evaluated on the test data set with added muscle artifact noise performed differently for both classes. The KNN took the lead by correctly identifying the true positive samples of the NOR class with a maximum recall of 0.61 and dropping to a min of 0.5 at SNR = −2 dB, followed by the RF and SVM, respectively, at SNR = 8 dB. However, the RF and SVM dropped significantly to 0.1 at SNR = −2 dB. Contrarily, for the MI class, the RF and SVM, as they took the lead, respectively, correctly identified more true positive samples of the MI class than the KNN, and the number of such samples increased significantly from nearly 0.79 as the quality of the signal decreased according to the SNR levels, achieving a maximum recall of 0.9 at SNR = −2 dB. In contrast to the other models, the KNN, which came second in performance, decreased accordingly with SNR levels by correctly identifying the true positive samples from 0.66 to a minimum recall of 0.58 at SNR = −2 dB. Similarly, in the case of the test data set with electrode motion artifact noise, the classifiers behaved in the same way for both classes, where the RF and SVM increased from a minimum recall of 0.73 at SNR = 16 dB to a maximum recall of 0.91 at SNR = −2 dB, respectively. However, for the test set with baseline wander, the models performed as expected by decreasing according to the SNR levels, where the KNN took the lead, followed by SVM and RF for the NOR class, achieving a minimum recall of nearly 0.34. Except at SNR = 2 dB, the KNN and SVM shifted back to increasing to a maximum recall of 0.55 and 0.40, respectively. The opposite performance was for the MI class, where the RF took the lead and did not decrease until the SNR was equal to 2 dB, followed by the SVM and KNN, with a minimum recall of 0.58 and 0.46, respectively.

We measured the F1-score of each model in predicting both classes in the test set with added noise; the results are presented in Figure 11. To identify both classes in all the noises added to the test set, all the classifiers behaved similarly, except at SNR = 16 dB, where the SVM had the highest F1-score in identifying the NOR class for the test set with muscle artifact noise, followed by the KNN and RF, and subsequently dropped to last place, as well as in the test set with baseline wander noise, where the RF and SVM had identically high F1-scores. At SNR = −2 dB, the SVM and RF models both dropped to approximately 0.2, and for the test set with the added baseline wander noise, to around 0.45. However, the KNN achieved the highest F1-score and maintained a decreasing performance above 0.5 for the rest of the SNR levels, predicting the NOR class. For the MI class, in all added noises to the test set, the RF took the lead, followed by the SVM and RF, achieving a high F1-score. RF and SVM classifiers maintained decreasing performances above 0.6, except for the baseline wander noise added to the test set, where it was above 0.55. On the other hand, the KNN decreased to values under 0.5, except for the test set with added electrode motion noise.

Finally, we measured the accuracy of each model in predicting both classes in each test set with added noise. Figure 12 illustrates the results of the accuracy achieved by each classifier. From the outset, we can observe that all models performed similarly in predicting both classes in each test data set with added noise, and it decreased accordingly with the SNR levels. The maximum accuracy achieved was around 0.67, and the minimum was around 0.47. However, at each level of SNR in each test set with added noise, all the models performed well at certain levels. At SNR = 16, the SVM took the lead, followed by RF and KNN, respectively, but it shifted between the SVM and the RF model, which took the lead at SNR = 8 dB, and from SNR = 2 to −2 dB, the KNN achieved the highest accuracy, followed by the RF and SVM, respectively. Similarly, in the test data set with added electrode motion noise, from SNR = 16 to 8 dB, the model behaved the same, where SVM took the lead, followed by RF and KNN. However, at SNR = 2 dB, the RF achieved high accuracy, followed by the KNN and SVM, with identical results, even though at SNR = −2 dB, the KNN took the lead, followed by the RF and SVM classifiers, with identical results. However, in the test set with added baseline wander, the RF model took the lead with high accuracy from SNR = 16 to 2 dB, but at SNR = −2 dB, the KNN achieved high accuracy, followed by the RF and SVM classifiers, with identical results.

## 5. Discussion

An automatic ECG myocardial infarction detection system should satisfy several requirements to be efficient in real-world practice, such as reliability and less complexity in hardware and software, especially concerning the number of resources required to run it, being more realistic in a clinical environment with high performance in decision-making. High performance in decision-making can be very tricky as the system or the model can provide efficient results as long as it is tested in the laboratory under perfect conditions, which makes it difficult to have the same efficiency in real-world practice, where there are a lot of variables and factors that can affect the performance of the system. In this case, in this paper, we evaluated the robustness of the automatic ECG myocardial infarction detection system under morphological changes in the data that could affect the behavior and efficiency of the classification system. This process of robustness characterizes how effective the model is while being tested on different (but similar) data sets; if the model can still keep a high-predicting performance as it is trained perfectly, then how much could it decrease as bad data are fitted to the model? Considering these requirements, we tested three well-known supervised machine learning models by training them on spotless data, filtered with no feature extractions, and evaluated their performances on the original data, the same data without filtering, and then on data with added real noise with different SNR levels, which means different qualities of the ECG signal. The real noises used are well known for their effects on ECG recording processes as they are related to the human body, such as the muscle (EMG) artifact, or the physical effects of the machine, such as an electrical effect, for instance, baseline wander or electrode motion artifacts. These effects can change the morphology of the ECG signal and lead to the miss-predicting of abnormalities by the classification model. Thus, a robust algorithm cannot degrade too much when it is tested on different data by either adding noise or using data without a filtering process. These experiments emulate approximate probable situations in real-world practice. According to our study results, which included two stages—(1) training and testing the classifiers on preprocessed data; and (2) testing the robustness of the classifiers in predicting the samples against a variety of noises—the RF model achieved high performance and outperformed the other approaches in correctly identifying more true positive samples of the MI class. This performance remains critical as it is less harmful to fail to predict a patient with normal conditions (NOR class) than to fail to predict a patient with myocardial infarction who could be at risk of dying from a heart attack. Moreover, it maintained this performance even when tested on the unfiltered test data set and, similarly, when tested against each noise added to the test data at all SNR levels Figure 10, as well as for the F1-score Figure 11. In addition, it achieved high precision for both classes at all SNR levels Figure 9. On the other hand, for the NOR class, the KNN achieved a high recall identically with SVM and decreased by 0.09, with a difference of 0.01 compared to SVM, which did not decrease much, Figure 6. However, the KNN outperformed the other models in correctly identifying more true positive samples of the NOR class when tested on each noise added to the test data set and maintained the performances at all SNR levels Figure 9, as well as in terms of the F1-score Figure 11. Overall, we can say that the RF model achieved good performance, especially in predicting the MI class, and the KNN in predicting the NOR class. Yet, the SVM model performed in the middle, according to all the metrics, which seems to indicate a balanced performance between the two classes. Furthermore, in terms of accuracy in predicting both classes, RF and SVM classifiers performed better by achieving high accuracy for all the noises added to the test data set and at nearly all the levels of SNR, but RF took the lead in most cases, surpassing SVM. The KNN stayed solid for the lower SNR levels, which meant a lower signal quality, where it achieved higher accuracy than other models. Overall, all the classifiers kept an accuracy level above 0.5, which is encouraging, keeping in mind the harmfulness of the noises that deteriorate the signal quality and change its features significantly. However, it is not very acceptable to have an accuracy level of under 0.5 for all the models, which is the case for the SNR levels from 2 dB to 2 dB for the muscle artifact noise. As for the electrode motion noise at SNR = −2 dB, the SVM and RF achieved accuracies of under 0.5. However, all the models can be considered sustainable models in specific cases for certain noises, as they maintained values above approximately 0.5 for all noise types presented in this study. For instance, RF may be preferred in the case of the baseline wander, SVM in the case of the electrode motion, and KNN for lower SNR levels (−2 dB) for all noise types. These results make the models pretty efficient, even with the lower quality of signals affected by different types of real noises. As a result, it can be thought of as a more realistic (in practice) and generic approach to dealing with scenarios in which ECG signals are affected by any type of noise, particularly the electrode motion artifact noise and muscle artifact noise, which are both frequent noises that have negative impacts on the ECG signal. On the other hand, to achieve high performance and enhance the prediction of particular categories and reduce wrongful predictions of some samples, which was not the aim of this study, perhaps it may be desirable to use the voting model or majority voting ensemble. This approach is preferred when there are two or more models that perform well in forecasting particular classes [23], as in the case of the results achieved in this study.

As far as we know, the models employed for inter-patient research for the classification of myocardial infarction have never been subjected to a robustness test, and there is no such approach for testing robustness precisely the way we used to evaluate the robustness of the model classification’s performance. Instead, we compared the performance achieved in this study with one paper that used an approximate approach of testing the robustness of an ECG model classification using an intra-patient paradigm and testing it against white noise, with other papers that worked on the same classification task using the inter-patient paradigm but with a different separation method and the same number of leads (12 leads). Overall accuracy achieved in this study seems to be non-competitive with other works, which was not the aim of this study, as shown in Table 8, especially with machine learning methods. Instead, it is acceptable, as the trained classifier achieved good accuracy (similar to the literature). Furthermore, it is encouraging in terms of robustness and accuracy dropping in comparison with reference [43], which used a deep learning classifier with defense methods on an intra-patient paradigm. However, it should be noted that the intra-patient paradigm is mostly focused on beat types, which implies that a certain person’s ECG recording may be found in both data sets (train and test data). Thus, it allows the model to strongly predict some classes because it learns all of the features of that sample. Thus, the results may indeed be biased, resulting in unfairly optimistic predictions [73]. Yet, for more realistic scenarios in real-world practice, the inter-patient paradigm is the most appropriate separation paradigm, which provides a reasonable indication of the model performance [23]. Therefore, the models investigated in this study have the potential to be highly competitive as robust models against the diversity of noise investigated in this study, especially as they were tested on an inter-patient paradigm. Furthermore, this study shows that the proposed robustness assessment test might be adequate to highlight some drawbacks of the models in predicting specific classes in certain circumstances, particularly against real noises, as tested in this study, since the ECG signal is vulnerable to noises. As a result, such a robustness assessment of the classifier’s performance could be considered before implementing such AI solutions (classifier models) in real-world clinical practices. Overall, all of the classifiers maintained good performances against different noise types tested in this study, even with non-competitive accuracy, compared to the literature, as shown in Table 8. Therefore, they were considered sustainable and robust models, as they had enough robustness against all the noises tested, especially as they achieved acceptable results in terms of low-quality signals, which did not drop extremely under 0.5.

## 6. Conclusions

In this study, we investigated an automatic ECG myocardial infarction detection system and presented a new approach to evaluate its robustness and durability performance in classifying the myocardial infarction with no feature extractions under different types of noise. We employed three well-known supervised machine learning models: support vector machine (SVM), k-nearest neighbor (KNN), and random forest (RF), and tested the performance and robustness of these techniques in classifying NOR and MI using real ECG records from the PTB database (after normalization and segmentation of the data) with a suggested inter-patient paradigm separation, in addition to noises from the MIT-BIH noise stress test database (NSTDB). Finally, we measured four metrics: accuracy, precision, recall, and F1-score. The simulation revealed that all the models performed well, with values of over 0.50 at lower SNR levels, in terms of all the metrics investigated against different types of noise, indicating that they are encouraging and acceptable under extreme noise situations and are considered sustainable and robust models for specific forms of noise. All the methods tested could be used as ECG myocardial infarction detection tools in real-world practice under challenging circumstances.

## Figures and Tables

**Figure 1 jcm-11-04935-f001:**
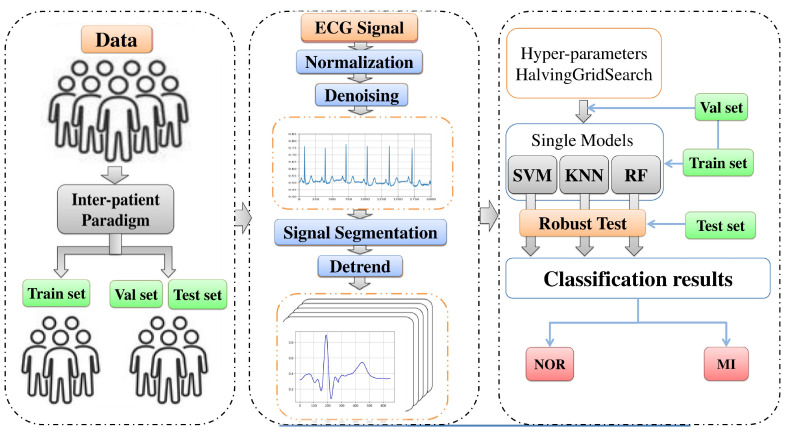
Overall procedures in ECG myocardial infarction classification based on the proposed models.

**Figure 2 jcm-11-04935-f002:**
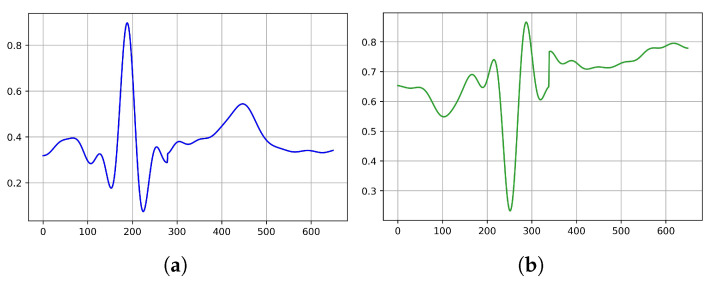
An illustration of two class samples after the preprocessing stage: (**a**) normal sinus rhythm (NOR) and (**b**) myocardial infarction (MI).

**Figure 3 jcm-11-04935-f003:**
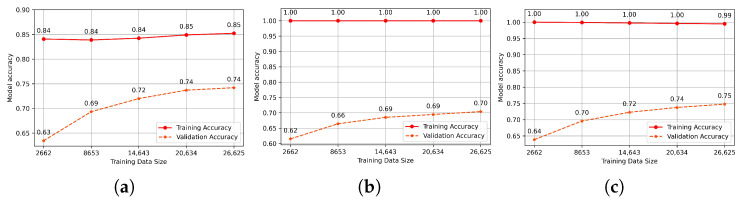
(**a**) SVM model learning curve, (**b**) KNN Model learning curve, (**c**) RF Model learning curve.

**Figure 4 jcm-11-04935-f004:**
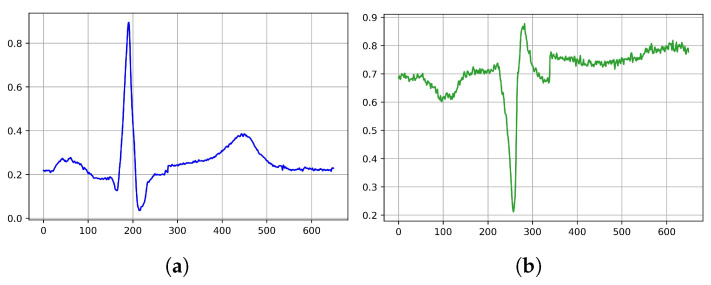
Illustration of original ECG signal: (**a**) normal sinus rhythm, (**b**) myocardial infarction sinus rhythm.

**Figure 5 jcm-11-04935-f005:**
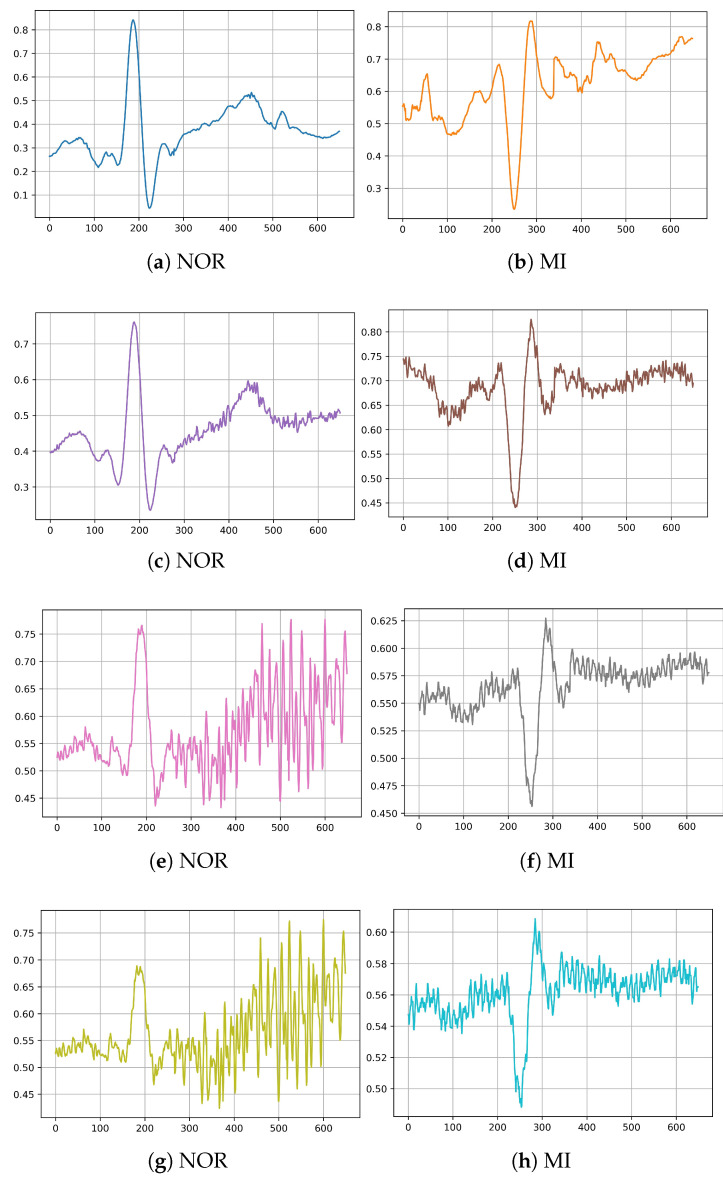
Illustration of different ECG signals after adding noises: (**a**,**b**) added electrode motion noise with SNR = 16 dB, (**c**,**d**) added baseline wander noise with SNR = 8 dB, (**e**,**f**) added muscle artifact noise with SNR = 2 dB, (**g**,**h**) added muscle artifact noise with SNR = −2 dB.

**Figure 6 jcm-11-04935-f006:**
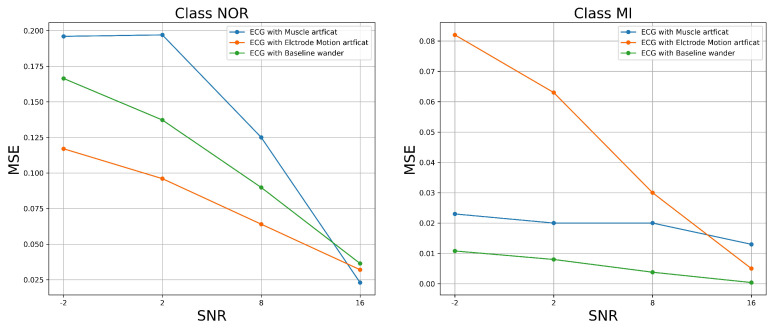
MSE versus SNR of the ECG-filtered signal with added noise.

**Figure 7 jcm-11-04935-f007:**
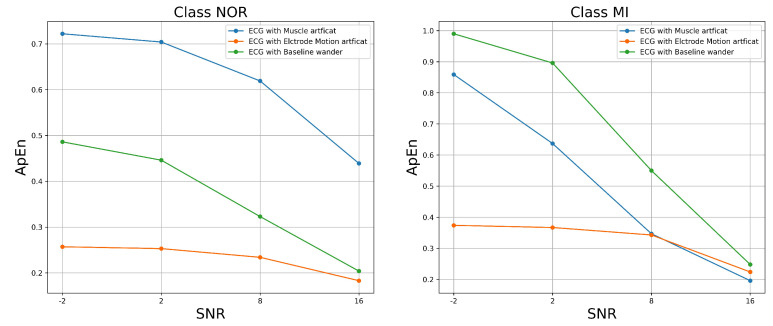
ApEn versus SNR of the ECG-filtered signal with added noise.

**Figure 8 jcm-11-04935-f008:**
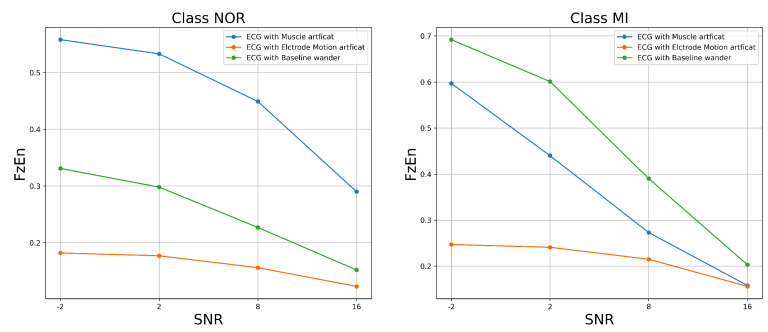
FzEn versus SNR of the ECG-filtered signal with added noise.

**Figure 9 jcm-11-04935-f009:**
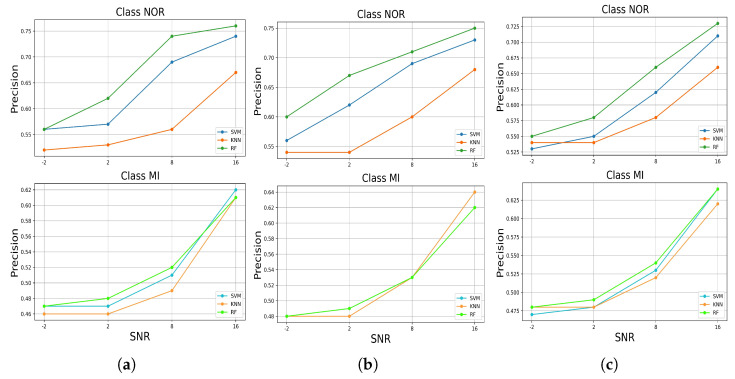
The precision performance of predicting the classes of the signal with added noises by the trained models, (**a**) for added muscle artifact noise, (**b**) for added electrode motion noise, (**c**) for added baseline wander noise.

**Figure 10 jcm-11-04935-f010:**
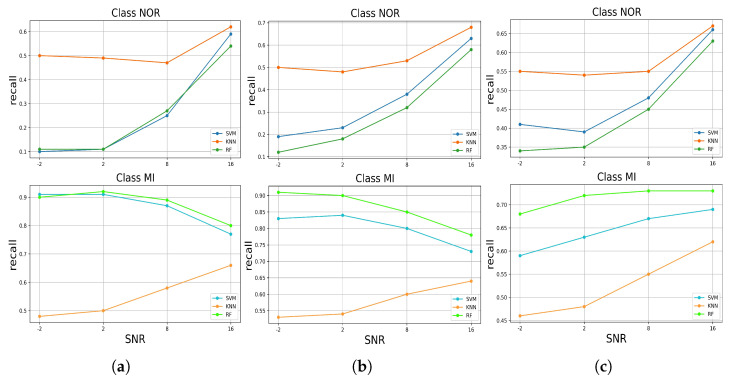
The recall performance of predicting the classes of the signal with added noises by the trained models, (**a**) for added muscle artifact noise, (**b**) for added electrode motion noise, (**c**) for added baseline wander noise.

**Figure 11 jcm-11-04935-f011:**
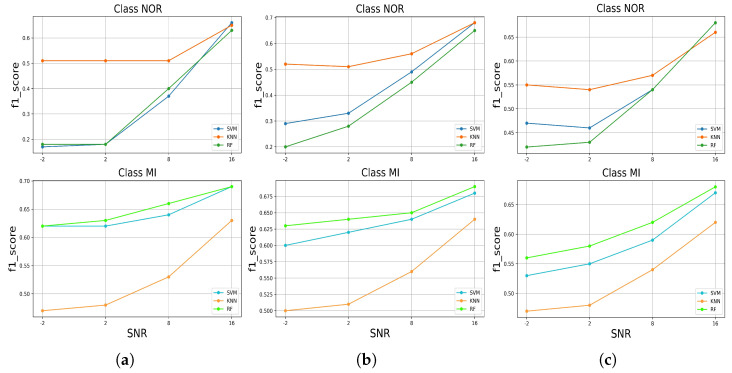
The F1-score performance of predicting the classes of the signals with added noises by the trained models, (**a**) for added muscle artifact noise, (**b**) for added electrode motion noise, (**c**) for added baseline wander noise.

**Figure 12 jcm-11-04935-f012:**
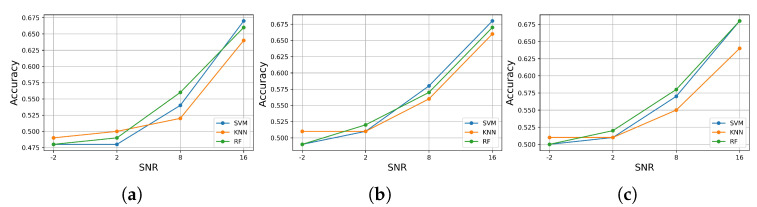
The accuracy performance of predicting the classes of the signal with added noises by the trained models, (**a**) for added muscle artifact noise, (**b**) for added electrode motion noise, (**c**) for added baseline wander noise.

**Table 1 jcm-11-04935-t001:** The group separation with the number of records in each group from the PTB database.

Data Set	Class	No. of Records	No. of 12-Lead Records
G1	NOR	27	324
	MI	27	324
G2	NOR	26	312
	MI	26	312
Total		106	1272

**Table 2 jcm-11-04935-t002:** Parameter selections.

Data Set	Class	No. of Samples
G1	NOR	13.380
	MI	13.245
G2	NOR	12.135
	MI	10.755

**Table 3 jcm-11-04935-t003:** Optimized hyperparameters for each model.

Models	Range of Grid
SVM	kernel = [‘rbf’, ‘sigmoid’], C = [10, 100], gammas = [0.1, 0.01, 0.001]
kNN	metric = [‘euclidean’, ‘manhattan’], n-neighbors = [1:11] interval: 2,
	weights = [‘uniform’, ‘distance’]
RF	n-estimators = [200, 400, 800], criterion = [‘gini’, ‘entropy’],
	max-depth = [5, 15, 25], min-samples-split = [5, 10, 15]

**Table 4 jcm-11-04935-t004:** Prediction performance comparison.

Classifier	Classes	*Precision*	*Recall*	*F1-Score*	*Accuracy*
SVM	NOR	0.75	0.78	0.76	0.74
	MI	0.73	0.70	0.72	
KNN	NOR	0.69	0.78	0.73	0.70
	MI	0.71	0.61	0.66	
RF	NOR	0.76	0.77	0.77	0.75
	MI	0.74	0.73	0.73	

**Table 5 jcm-11-04935-t005:** Classification report.

Classifier	Classification Report			
SVM			predicted label	
			NOR	MI
	True label	NOR	0.78	0.22
		MI	0.3	0.70
			predicted label	
			NOR	MI
KNN	True label	NOR	0.78	0.22
		MI	0.39	0.61
			predicted label	
			NOR	MI
RF	True label	NOR	0.77	0.23
		MI	0.27	0.73

**Table 6 jcm-11-04935-t006:** Prediction performance comparison.

Classifier	Classes	*Precision*	*Recall*	*F1-Score*	*Accuracy*
SVM	NOR	0.70	0.70	0.70	0.68
	MI	0.66	0.66	0.66	
KNN	NOR	0.66	0.69	0.67	0.65
	MI	0.63	0.60	0.61	
RF	NOR	0.72	0.61	0.66	0.66
	MI	0.62	0.73	0.67	

**Table 7 jcm-11-04935-t007:** Classification report.

Classifier	Classification Report			
SVM			predicted label	
			NOR	MI
	True label	NOR	0.70	0.30
		MI	0.34	0.66
			predicted label	
			NOR	MI
KNN	True label	NOR	0.69	0.31
		MI	0.40	0.60
			predicted label	
			NOR	MI
RF	True label	NOR	0.61	0.39
		MI	0.27	0.73

**Table 8 jcm-11-04935-t008:** Comparison with other works in the literature.

Ref.	FeEx	CM	Acc(%)	Se		Sp	
Dohare et al. [38]	yes	SVM	96.66	96.66		96.66	
Sopic et al. [27]	yes	RF	82.36	87.95		78.82	
Diker et al. [39]	yes	SVM	87.80	86.97		88.67	
Liu et al. [5]	yes	CNN + BLSTM	93.08	94.42		86.29	
Han and Shi [41]	yes	ML-ResNet	95.49	94.85		97.37	
Wang et al. [26]	yes	KNN	77.51	73		82.01	
Ma and Liang [43]	No	CNN + 1.0NSR	0.65–0.83 *	-		-	
				NOR		MI	
				**Se**	${\mathrm{Pe}}^{+}$	**Se**	${\mathrm{Pe}}^{+}$
Proposed method	No	RF	0.5–75	77	76	73	74

* Intra-patient paradigm and against white noise.

## Data Availability

Not applicable.

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
