# Peer review of "A Robustness Evaluation of Machine Learning Algorithms for ECG Myocardial Infarction Detection"

_jcm, 2022, doi:10.3390/jcm11174935_

Round 1

Reviewer 1 Report

The paper presents a comparative study using three machine learning classifiers for myocardial infarction detection from noisy ECGs.

The novelty of the paper should be more clearly emphasized. A more comprehensive review and discussion of methods investigating robustness of the studied machine learning models may be included. 

It would be interesting if the authors could propose methods to improve the classification performance in the presence of noisy data.

From a clinical point of view, it will be interesting to develop methods which can identify subendocardial infarction and other particular cases such as infarction in patients with left buddle branch block. In these situations, ECGs are not enough for a correct diagnosis.  Clinicians need to determine biological markers of myocardial necrosis.

Minor corrections:

- Missing reference on line 68

- typo on line 189: “prposed” should be replaced with proposed

Reviewer 2 Report

Line 1: infarction and not infraction

Line 9: the first time that you use terms NOR and MI plesase specify in brackets what do they mean (normal and Myocardial Infarction) because NOR is not an habitually used acronym

The research compares three different infarction detection systems particularly useful in automatic diagnosis of myocardial infarction under challenging circumstances, like noisy ecgs.

Topic is not original, but to my knowledge there aren't other papers that compare these three methods.

The paper is well written but sincerely boring, not very easy to read.

It is not clear why they entitle the manuscript "algorithms for ECG arrhythmia detection" and then they speak about infarction. Arrhythmia and infarction are different topics. Anyway conclusions are consistent with the arguments presented.
